

# Should I use fixed effects or random effects when I have fewer than five levels of a grouping factor in a mixed-effects model?

Dylan G.E. Gomes[1,2]

[1] Biological Sciences, Boise State University, Boise, Idaho, United States
[2] Cooperative Institute for Marine Resources Studies, Hatfield Marine Science Center, Oregon State University, Newport, Oregon, United States

## ABSTRACT

As linear mixed-effects models (LMMs) have become a widespread tool in ecology, the need to guide the use of such tools is increasingly important. One common guideline is that one needs at least five levels of the grouping variable associated with a random effect. Having so few levels makes the estimation of the variance of random effects terms (such as ecological sites, individuals, or populations) difficult, but it need not muddy one's ability to estimate fixed effects terms—which are often of primary interest in ecology. Here, I simulate datasets and fit simple models to show that having few random effects levels does not strongly influence the parameter estimates or uncertainty around those estimates for fixed effects terms—at least in the case presented here. Instead, the coverage probability of fixed effects estimates is sample size dependent. LMMs including low-level random effects terms may come at the expense of increased singular fits, but this did not appear to influence coverage probability or RMSE, except in low sample size ($N = 30$) scenarios. Thus, it may be acceptable to use fewer than five levels of random effects if one is not interested in making inferences about the random effects terms (*i.e.* when they are 'nuisance' parameters used to group non-independent data), but further work is needed to explore alternative scenarios. Given the widespread accessibility of LMMs in ecology and evolution, future simulation studies and further assessments of these statistical methods are necessary to understand the consequences both of violating and of routinely following simple guidelines.

Corresponding author
Dylan G.E. Gomes,
dylangomes@u.boisestate.edu

## INTRODUCTION

As ecological datasets are inherently noisy and as researchers gain increased access to large datasets, statistical analyses in ecology are becoming more complex (*Low-Décarie, Chivers & Granados, 2014*). Yet, advances in computing power and freely available statistical software are increasing the accessibility of such analyses to non-statisticians (*Patil, Huard & Fonnesbeck, 2010*; *Gabry & Goodrich, 2016*; *Salvatier, Wiecki & Fonnesbeck, 2016*; *Brooks et al., 2017*; *Bürkner, 2017*; *Carpenter et al., 2017*; *Rue et al., 2017*). As these methods have become more complex and accessible to ecologists, fisheries and wildlife managers,

and evolutionary biologists, the need to guide the use of such tools is becoming increasingly important (*Bolker, 2008*; *Bolker et al., 2009*; *Zuur, Ieno & Elphick, 2010*; *Kéry & Royle, 2015*; *Kass et al., 2016*; *Zuur & Ieno, 2016*; *Harrison et al., 2018*; *Arnqvist, 2020*; *Silk, Harrison & Hodgson, 2020*). Generalized linear mixed-effects models (GLMM), for example, have become a widespread tool that allows one to account for correlation between data that come from non-independent blocks or populations (*i.e.* random effects; also known as varying effects) (*Bolker, 2008*; *Kéry & Royle, 2015*; *Powell & Gale, 2015*; *Harrison et al., 2018*; *McElreath, 2020*).

Generalized linear mixed-effects models are flexible in that they can handle a variety of response distributions such as binomial (*e.g.* presence/absence of a species, alive/dead individuals) and negative binomial (*e.g.* wildlife counts; *Warton, 2005*). When the response distribution is Gaussian (also known as normal; *e.g.* body length, vocalization frequency), this is a special case of a GLMM that is referred to as a linear mixed-effects model (LMM). GLMMs (and LMMs) differ from their simpler counterparts, (generalized) linear models (GLMs and LMs), in that they include random effects, in addition to the fixed effects (hence mixed-effects).

Fixed effects are also often called predictors, covariates, explanatory or independent variables in ecology. Fixed effects can include both variables of interest (*e.g.* average temperature in climate change studies or sound pressure levels in anthropogenic noise research) and other "nuisance" variables that are only included to control for unexplained variation but are not directly useful to understanding the research question at hand (*e.g.* size, age or sex of an animal, or % riparian vegetation or elevation at a sampling site).

Random effects allow one to deal with spatiotemporal autocorrelation, use partial pooling to borrow strength from other populations, account for grouping or blocked designs (*e.g.* repeated-measures data from sites or individuals), and estimate population-level parameters, among other applications (*Kéry & Royle, 2015*). Thus, the experimental design (*e.g.* grouping blocks) often determines the random effects structure in a model (*Barr et al., 2013*; *Arnqvist, 2020*). Random effects are assumed to be drawn *randomly* from a distribution—usually a Gaussian distribution—during the data-generating process (note that the response distribution need not also be Gaussian). The following equation includes both fixed and random effects;

$$y_i = \alpha_{j(i)} + \beta_1 X_{1_i} + \beta_2 X_{2_i} + \varepsilon_i; \tag{1}$$

$$\alpha_j \sim Normal(\mu, \sigma^2), \tag{2}$$

where $\alpha_{j(i)}$ represents the random effect group $j$ (*e.g.* site or individual) to which observation $i$ belongs, which moves the group $j$ intercept up or down from the overall intercept ($\mu$; mean of all groups). $\beta_1$ and $\beta_2$ are the slope parameters (*i.e.* fixed effects) for two generic predictor variables, which vary by observation $i$ ($X_{1_i}$ and $X_{2_i}$ respectively). The error term $\varepsilon_i$, is unique to each observation $i$, and is often thought to come from a normal distribution (as does $\alpha_j$ in Eq. (2)).
**Table 1 Advantages of estimating grouping factors as fixed (LM) vs random (LMM) effects.** This table describes some advantages of fitting linear models (LMs) vs linear mixed-effects models (LMMs), when a grouping factor can be estimated as either a fixed effect or random effect, respectively.

| LM | LMM |
|---|---|
| Faster to compute | Can provide increased precision |
| Simpler to use (less prone to user error) | Can incorporate hierarchical grouping of data |
| Conceptually simpler, such that metrics like $R^2$ are easier to compute | Are (often) conceptually correct models of the system |
| Avoid concerns about singular fits | Can share information across groups (partial pooling), which aids in the estimation of groups with few observations |
| Preclude users from making inappropriate generalizations to unobserved levels of the grouping variable | Allows generalization to unobserved levels of the grouping variable |
| Avoid assumptions about the distribution of the random effects | In a study with multiple analyses with varying numbers of grouping variable levels, it would be more consistent to use LMMs throughout rather than switching to LMs at some threshold value |
| Estimating the number of degrees of freedom used is straightforward | Random effects use fewer degrees of freedom |

Fixed effects, on the other hand, rely on a series of indicator variables that are either turned on (*i.e.* $Site_{1(i)} = 1$) or off (*i.e.* $Site_{1(i)} = 0$), which allows for the estimation of the effects of each grouping level (*e.g.* $\alpha_1, \alpha_2, \ldots, \alpha_n$) to occur independently, as in

$$y_i = \beta_1 X_{1_i} + \beta_2 X_{2_i} + \alpha_1 Site_{1(i)} + \alpha_2 Site_{2(i)} + \ldots \alpha_n Site_{n(i)} + \varepsilon_i, \tag{3}$$

One obvious advantage of using random effects that follows is that fewer terms need to be estimated (only $\mu, \sigma^2$ for random effects and $\alpha_1, \alpha_2, \ldots, \alpha_n$ for fixed effects; see Table 1 for other benefits of using fixed effects or random effects). Another advantage is in how they are estimated. Fixed effects are estimated using maximum likelihood while random effects are estimated with partial pooling [also called "Best Linear Unbiased Prediction" (BLUP)] (*Robinson, 1991*; *Gelman, 2005*). Partial pooling allows different groups (*i.e.* levels of the random effect) to share information, as they are considered to be not completely independent, such that groups with few observations are more easily estimated than they would be if estimated as completely independent groups (this latter case would be considered "no pooling", which is what happens when these groups are estimated as fixed effects). The result is that group estimates are pulled toward the overall mean of all groups (*i.e.* the distribution mean that the groups are drawn from; Eq. (2))—also known as shrinkage. This also means that predictions can more easily be generalized to groups that were not measured (assuming those unmeasured groups come from the same population as the measured groups). As a note, the random effects estimator is the same as the fixed effects estimator when the variance is infinite (*Gelman & Hill, 2006*). That is, when $\sigma^2$ becomes $\infty$ as in $\alpha_j \sim Normal(\mu, \infty)$, each $\alpha$ is considered independent, as is the case in a fixed-effects only model (*Gelman & Hill, 2006*; *Clark & Linzer, 2015*). For completeness, a "full pooling" model would be one in which $\sigma^2 = 0$; $\alpha_j \sim Normal(\mu, 0)$, or in other words, each group has the same effect.

If we are interested in the variability of a random effect (*e.g.* individuals, sites, or populations; $\sigma^2$ in Eq. (2)), it is difficult to estimate this variation with too few levels of individuals, sites, or populations (*i.e.* random effects terms).

"When the number of groups is small (less than five, say), there is typically not enough information to accurately estimate group-level variation" (*Gelman & Hill, 2006*).

"…if interest lies in measuring the variation among random effects, a certain number is required…To obtain an adequate estimate of the among-population heterogeneity—that is, the variance parameter—at least 5–10 populations might be required" (*Kéry & Royle, 2015*).

"With <5 levels, the mixed model may not be able to estimate the among-population variance accurately" (*Harrison et al., 2018*).

"Strive to have a reasonable number of levels (at the very least, say, four to five subjects) of your random effects within each group" (*Arnqvist, 2020*).

The guideline that random effects terms should have at least five levels (*i.e.* blocks) is backed by only limited documented empirical evidence (*Harrison, 2015*), while recent work suggests that mixed-models can correctly estimate the variance with only two levels in some cases (*Oberpriller, de Souza Leite & Pichler, 2021*). Yet it is intuitive that too few draws from a distribution will hinder one's ability to estimate the variance of that distribution. Indeed, in each of the above segments of quoted text, the authors suggest that at least five levels are needed for *estimation of group-level, or among-population, variance*. However, it is my observation that there is some confusion about this rule and it is often adhered to out of context, where authors or reviewers of ecological journals suggest that one cannot use random effects terms if they do not contain at least five levels (*Bain, Johnson & Jones, 2019*; *Bussmann & Burkhardt-Holm, 2020*; *Evans & Gawlik, 2020*; *Gomes & Goerlitz, 2020*; *Zhao, Johnson-Bice & Roth, 2021*), although others are aware that this rule exists yet ignore it (*Latta et al., 2018*; *Fugère, Lostchuck & Chapman, 2020*; *Gomes, Appel & Barber, 2020*; *Allen et al., 2021*). For context, within a small sample of the most recent articles in the journal *Ecology* (N = 50), 18 articles mentioned 29 different random effects—only one of which (~3%) included fewer than five levels (Fig. 1; Table S1).

Simulations by *Harrison (2015)* demonstrate that estimates of random effect variances can be biased more strongly when the levels of random effects terms are low, yet in this work it appears that slope ($\beta$) estimates for fixed effects terms are generally not more biased with only three random effects levels compared to five. Similarly, in a recent pre-print *Oberpriller, de Souza Leite & Pichler (2021)* found that type I error and coverage of fixed effects estimates in mixed models stayed constant across different numbers of levels of a grouping variable (even when there were only two groups). In many cases, the variance of the random effect is not of direct ecological interest.
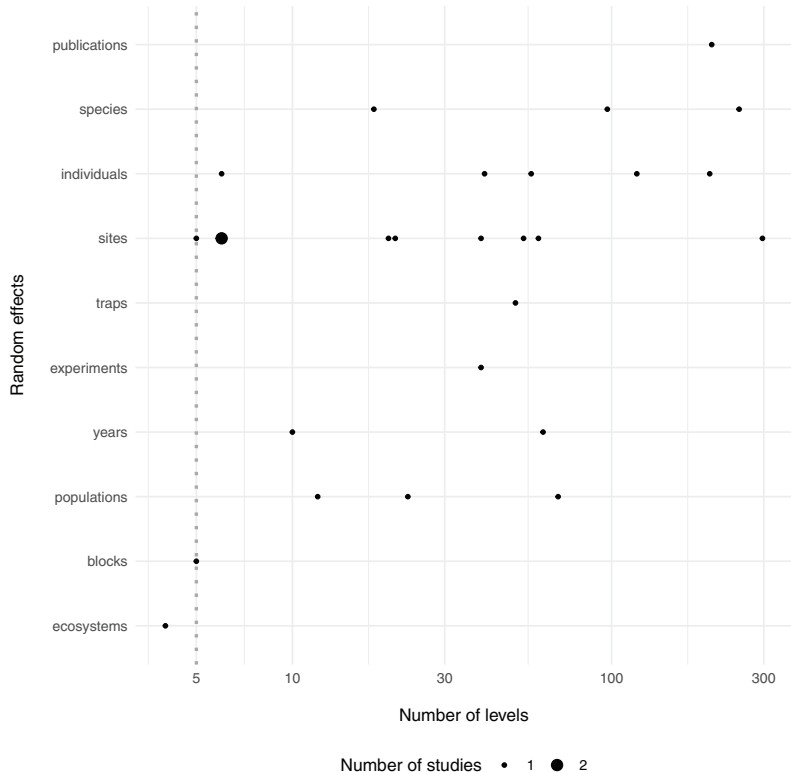

**Figure 1 Number of levels of random effects terms from 50 recent papers in Ecology.** The type (*y* axis) and number of levels (*x* axis) of random effect terms are displayed from a survey of the 50 most recent papers in the journal Ecology. Dotted line at 5 levels, shows a common cutoff for estimation of grouping factors as random effects. Twenty-nine random effects were mentioned in 18 papers, although it was unclear how many levels two random effects had, so they were omitted here (see Table S1).

"…in the vast majority of examples of random-effects (or mixed) models in ecology, the random effects do *not* have a clear ecological interpretation. Rather, they are merely abstract constructs invoked to explain the fact that some measurements are more similar to each other than others are—*i.e.*, to model correlations in the observed data" (*Kéry & Royle, 2015*).

Thus, it can be unclear whether or not it is appropriate to use random effects when there are fewer than five levels of the grouping variable of a random effect in situations where one does not directly care about the nuisance among-population variance, but instead is interested in estimates and uncertainty of fixed effects parameter estimates. The current state of practice in ecology, when the grouping variable has too few levels, is to treat that grouping variable as a set of fixed effects rather than a random effect. Yet there is only limited work (see above) exploring how this choice affects parameter estimates (*Clark & Linzer, 2015*), especially with fewer than five levels of a random effect (*Oberpriller, de Souza Leite & Pichler, 2021*). I question whether this rule can be safely ignored in some cases (*i.e.* when one does not care about among-population variance).

Here, I simulate unbalanced datasets (*i.e.* unequal numbers of observations per level of the grouping factor) to assess whether *fixed effects* estimates are more biased when the grouping factor consist of fewer than five levels. I assess RMSE and coverage probability between LMs and LMMs that only differ in the grouping factor (specified as a fixed and random effect, respectively), and the number of singular fits for LMMs with varying levels of the random effect term.

## METHODOLOGY

All simulation of datasets and model fitting was done in R v4.0.4 (*R Core Team, 2017*) with functions lmer (LMM) or lm (LM) in the package lme4 (version 1.1–27.1) or in stats, respectively (*Bates et al., 2015b*; *R Core Team, 2017*). All visualizations were accomplished with the aid of the R package ggplot2 (*Wickham, 2011*).

### Data generation

I modified code from *Harrison (2015)*, to explore the importance of varying two parameters in a linear mixed-effect model (LMM): the total number of observations in a dataset ($N$ = 30, 60, or 120), and the number of levels of the random intercept term (3, 5, or 10) in an unbalanced design. We can think of the latter as the number of individuals in an experiment or the number of field sites in a study. To simulate an unbalanced design, I first generated one observation for each level of the random intercept term (so that datasets had 3, 5, or 10 levels). I then randomly selected levels for the other observations until the total number of observations for the dataset was reached (30, 60, or 120). If any datasets were, by chance, balanced (*i.e.* equal numbers of observations for each level of the random effect), this dataset was discarded and the process was repeated. This served two purposes. Firstly, ecological datasets are often not perfectly balanced due to logistical, financial, and other constraints. Secondly, and importantly, in the specific case of a balanced design the fixed-effects parameter estimators are independent of all the random effects (*Wood, 2017*). The response variable $y_i$ was then generated with Eq. (1), which is reproduced here as an ecological example:

$$\text{tree height}_i = 73 + Site_{j(i)} + 2 \cdot sunlight_i + 0.2 \cdot carnivores_i + \varepsilon_i \tag{4}$$

$$Site_j \sim Normal(0, \sigma^2) \tag{5}$$

where the overall intercept, has been arbitrarily set to 73 for this example and $Site_{j(i)}$ represents the random effect (*e.g.* site or individual) $j$ to which observation $i$ belongs, which moves the group intercept up or down. That is, some sites have shorter trees on average and other sites have taller trees on average, but the entire forest (where all sites were randomly selected from; Eqs. (4) and (5)) has a mean height of 73. Thus, each observation within a site shared an intercept (the overall intercept plus $Site_{j(i)}$). $Site_{j(i)}$ was drawn from a normal distribution with mean = 0 and variance ($\sigma^2$) = 1. $X_{1_i}$ and $X_{2_i}$ (Eq. (1); sunlight and carnivores in Eq. (4)) were both randomly generated from a normal distribution with $\mu = 0$ and $\sigma^2 = 0.5$, which mimics standardized variables that are centered by their mean and scaled by two standard deviations (*Gelman, 2008*). While $X_{1_i}$ and $X_{2_i}$

were drawn from a normal distribution during data generation, their associated parameters $\beta_1$ and $\beta_2$ are not (these are the *fixed* effects). For all simulated datasets, parameter values were fixed at $\beta_1 = 2$ and $\beta_2 = 0.2$. In this ecological example, we might expect sunlight to have a stronger positive relationship ($\beta_1 = 2$) with tree height and the number of carnivores to have a weaker positive relationship ($\beta_2 = 0.2$) with tree height (*e.g.* a trophic cascade). The error term $\varepsilon_i$, unique to each observation $i$, is drawn from a normal distribution with $\mu_\varepsilon = 0$ and $\sigma^2_\varepsilon = 1$ (the same as Eqs. (2) and (4) above); this term is simply adding noise to our system during the data generating process, but is estimated implicitly as residual variance in many generic LMM functions.

## Model fitting simulations

For each of the nine combinations of scenarios (30, 60, or 120 observations by 3, 5, or 10 sites), I simulated 10,000 datasets. Each dataset was fit with a linear mixed-effect model (LMM with restricted maximum likelihood; REML) and a linear model (LM).

```
#LMM:
m1 <- lmer(y ~ x₁ + x₂ + (1|Site), data = dat)
R Code
```

where $x_1$ and $x_2$ are fixed effects (see Eq. (1)), and (1|Site) is the syntax for specifying a random intercept only ($\alpha_{j(i)}$ in Eq. (1)). In ecology, we often specify independent sites as unique levels of a random effect, so I use site here for demonstration purposes. But site can be replaced with individual, population, *etc.* Often the recommendation is to specify the random effects as fixed effects (LMM becomes LM) if one has fewer than 5 levels of random effects terms ($n < 5$ in $\alpha_{j(i)}$), specified in R as:

```
#LM:
m2 <- lm(y ~ x₁ + x₂ + Site, data = dat)
R Code
```

and mathematically defined as:

$$y_i = \beta_1 X_{1_i} + \beta_2 X_{2_i} + \alpha_1 Site_{1(i)} + \alpha_2 Site_{2(i)} + \ldots \alpha_n Site_{n(i)} + \varepsilon_i \qquad (6)$$

Now a $\alpha$ parameter is estimated for each site (or population) level independently. Site effects no longer come from a normal distribution (as in Eq. (2)), but each site still has its own intercept (*e.g.* $\alpha_1 Site_{1(i)}$ for site 1 and $\alpha_2 Site_{2(i)}$ for site 2). Thus, both a LMM and a LM were fit to each simulated dataset (n = 10,000) of each of the nine combinations of data-generation (30, 60, or 120 observations by 3, 5, or 10 sites; for completeness: 90,000 total simulated datasets and 2 models fit to each dataset). This allowed for a comparison of the precision and accuracy of fixed effect slope estimation by LMMs and LMs, the latter fits the blocked structure of data (*i.e.* site-level grouping) as a fixed effect, rather than a random effect.

## Web of science search

On July 5, 2021, I used Web of Science to search all "Article" document types from the journal Ecology ("SO=ECOLOGY") from the current year (Timespan: Year to date). This resulted in 132 articles, of which I selected the most recent 50 articles (see data and

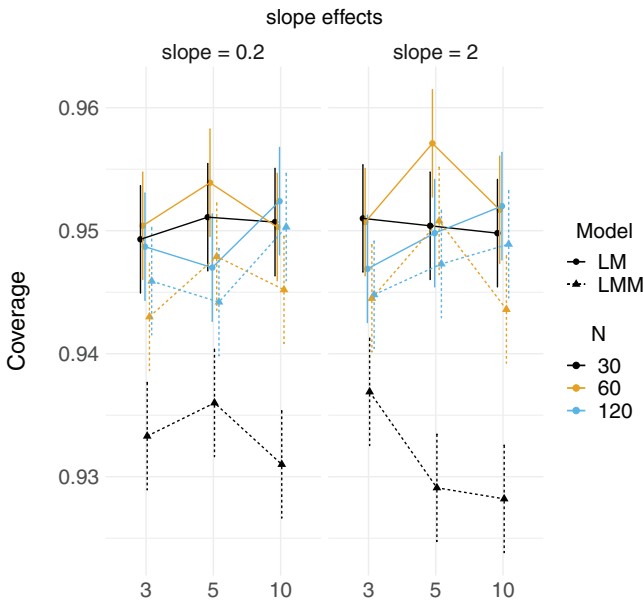

**Figure 2 95% Coverage probability of fixed effects in LMs and LMMs.** The proportion of simulation runs in which the true value was found within the 95% confidence intervals of model estimates is displayed on the $y$ axis, while the number of levels of a grouping factor is on the $x$ axis. The symbols represent linear models (LM) *vs* linear mixed-effect models (LMM) where the grouping factor is specified as a fixed effect *vs* a random effect, respectively, within the same datasets. Colors denote overall sample size ($N = 30$, 60, or 120). The left column shows data for the relatively weaker slope (0.2), and the right column shows data for the relatively stronger slope (2).

code statement). I then scanned the methods of each article for evidence of use of (generalized) linear mixed-effects models (LMMs or GLMMs) by searching the terms: lm, lmm, glm, glmm, linear, random, fixed, effects, mixed, model, and group within the entire text and by reading the text in the methodology sections.

# RESULTS

## Estimating model parameters and uncertainty

Linear mixed models and linear models were able to reconstruct simulated fixed effect relationships with no noticeable patterns in bias, regardless of number of levels of random effects or sample size. That is, both mean model parameter estimates ($\beta_1$ and $\beta_2$) were close to their true values (2 and 0.2, respectively; Fig. S1). The proportion of simulated runs within which the true fixed effect value was found within the 95% confidence intervals of these estimates was generally around the expected 0.95, but some LMM scenarios did not reach this value (which had somewhat lower coverage, in general, than LMs; Fig. 2). This was particularly the case when the total sample size was only 30 observations (black triangles in Fig. 2), but this did not appear to depend strongly on the number of levels of the random effect.

Relative root-mean-square error (RMSE) of fixed effect estimates was highest for lower sample sizes ($N = 30$; Fig. 3). RMSE does not appear to differ substantially between LM and

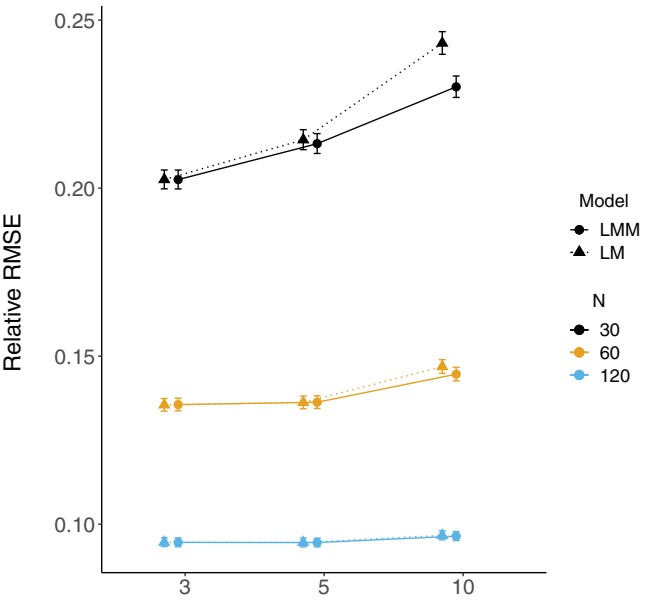

**Figure 3 Relative RMSE of model fixed effects estimates.** The relative RMSE of model fixed effects estimates is plotted against the number of levels of a grouping factor. Triangles represent linear model RMSE values (dotted line) and circles represent linear mixed-effect model RMSE values (solid line) for the slope = 2 estimates (see Fig. S2 for a plot of slope = 0.2), which is qualitatively similar. Colors denote simulated dataset sample size ($N$ = 30, 60, or 120).

LMMs and across the number of levels of random effects terms, although these mild differences are largest when sample sizes are lowest and the number of levels of the random effect is highest ($N$ = 30; $N$ levels = 10; Fig. 3, S2).

All LMM estimates of the random effects variance ($\sigma^2$) were not centered at the true value (*i.e.* under-estimates; Fig. 4). Estimates were further from true values with fewer levels of random effects and smaller sample sizes, although there was considerable overlap of confidence intervals with true values (Fig. 4).

There were more singular fits with fewer levels of random effects terms and with smaller sample sizes (Fig. 5). In most cases coverage did not differ between models with singular and non-singular fits, although differences were largest when sample sizes were lowest ($N$ = 30) and the number of levels of the random effect were highest ($N$ levels = 10; Fig. 6). Similarly, RMSE did not differ between models with singular and non-singular fits, except for the lowest sample size ($N$ = 30), highest number of levels of the random effect ($N$ levels = 10) scenario (Fig. 7).

## DISCUSSION

The simulations presented here demonstrate that fixed effects estimates are not necessarily more biased in LMMs vs LMs when there are fewer than five levels of a grouping factor, but population-level variance estimates (random effect variance) can be. When sample sizes were lowest, the coverage of fixed effects slope estimates is higher for LMs (relative to LMMs), but there were no consistent patterns for the number of levels of random effects

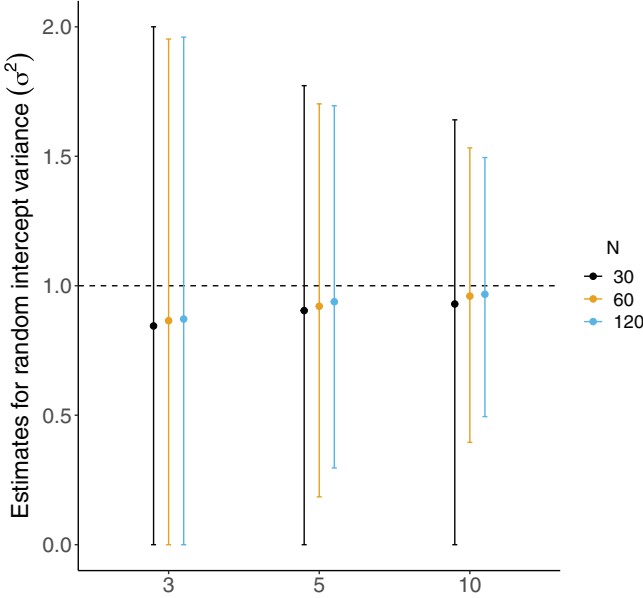

**Figure 4 Random effects variance estimates from LMMs of simulated data.** Each point is the mean point estimate for 10,000 simulated runs, whereas error bars are the 95% intervals (0.025 and 0.975 quartiles) of the distribution of 10,000 point estimates for the random effects variance. $N$ = the number of observations (*i.e.* number of rows) in each dataset. Dashed lines indicate the true value.

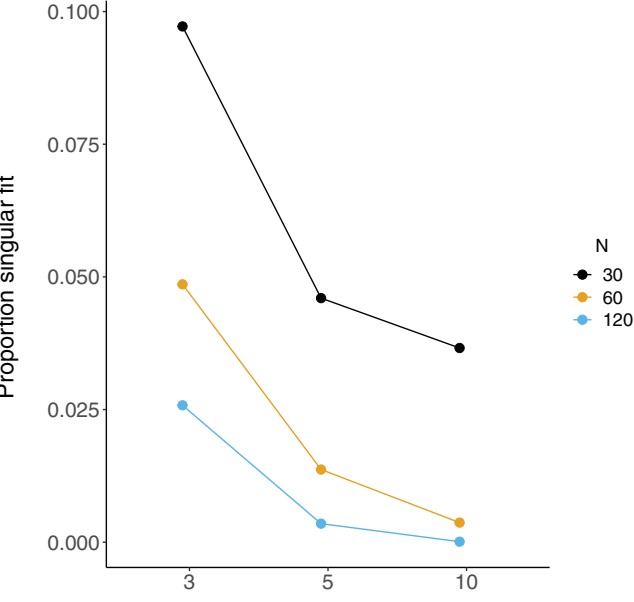

**Figure 5 Singular fits in LMM.** Each point is the proportion of linear mixed-effect model runs (total = 10,000) that had a singular fit (variances of one or more effects are zero, or close to zero). $N$ = the number of observations (*i.e.* number of rows) in each simulated dataset.

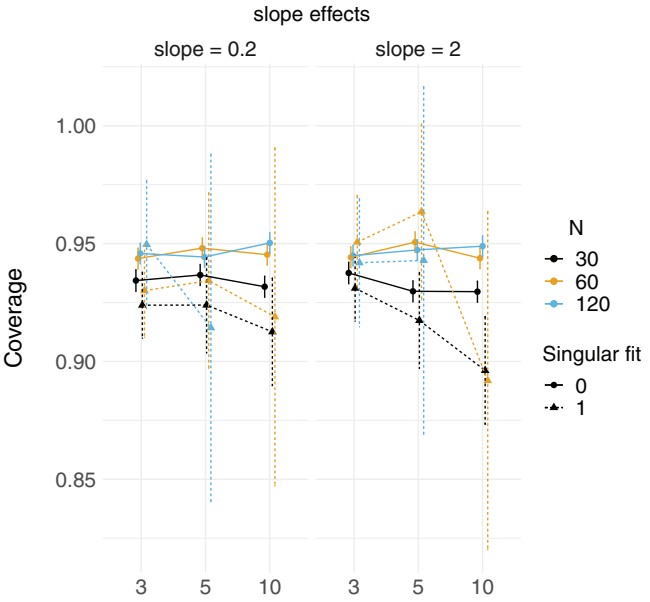

**Figure 6  95% Coverage probability of LMMs with singular fits.** The proportion of simulation runs in which the true value was found within the 95% confidence intervals of model estimates is displayed on the $y$ axis, while the number of levels of a grouping factor is on the $x$ axis. The symbols represent linear mixed-effect models (LMM) with a singular fit (variances of one or more effects are zero, or close to zero; singular = 1) or a non-singular fit (singular = 0). Colors denote overall sample size ($N$ = 30, 60, or 120). The left column is where the slope is relatively weaker (slope = 0.2), and the right column is where the slope is relatively stronger (slope = 2). Note that there was only one singular fit (out of 10,000 runs) for the largest data set ($N$ = 120) with the most levels of the random effect ($N$ levels = 10). Since this single point can only be 0 or 1, and is unlikely to reflect a mean value (that more simulation runs might elucidate), it was omitted here.               

terms, further suggesting that fixed effects estimates may be relatively robust when there are few (*i.e.* < 5) levels of a random effect.

Fixed effects parameter estimation here does not appear to be strongly influenced by, nor biased by, the number of levels of random effects terms. Instead, uncertainty in those estimates was much more strongly influenced by sample size. While this pattern may appear to contradict the decreased uncertainty (with more random effects levels) around beta estimates in Fig. 2 of *Harrison (2015)*, this instead is likely due to differences in the way that sample size relates to the number of random effects levels. *Harrison (2015)* coded each random effect level to be associated with a fixed number of observations ($N$ = 20), such that each additional random effect level yielded an increased sample size. However, in the simulations here, sample size (*i.e.* number of observations) has been separated from the number of random effects terms (*e.g.* sites or individuals).

Despite these differences in coding, the estimation of random effects terms suggest consistent patterns with *Harrison (2015)* in that variance ($\sigma^2$) is more biased with fewer levels. This seems to support previous suggestions and simulations suggesting that few levels of random effects terms can make estimation of population-level variance difficult (*Gelman & Hill, 2006*; *Harrison, 2015*; *Kéry & Royle, 2015*; *Harrison et al., 2018*), but a

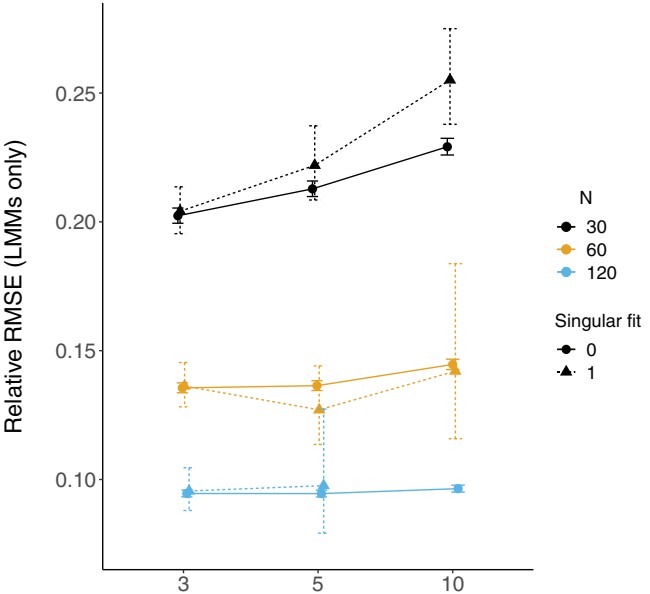

**Figure 7 Relative RMSE of LMM fixed effects estimates with singular fits.** The relative RMSE of model estimates is plotted on the *y* axis, while the number of levels of a grouping factor is on the *x* axis. The symbols represent linear mixed-effect models (LMM) with a singular fit (variances of one or more effects are zero, or close to zero; triangles; singular = 1) or a non-singular fit (circles; singular = 0). Colors denote overall sample size (*N* = 30, 60, or 120). The left column is where the slope is relatively weaker (slope = 0.2), and the right column is where the slope is relatively stronger (slope = 2). Note that there was only one singular fit (out of 10,000 runs) for the largest data set (*N* = 120) with the most levels of the random effect (*N* levels = 10). Since this single point may not reflect a mean value (that more simulation runs might elucidate), it was omitted here.

hard cutoff at five random effects levels remains arbitrary. The combination of these results suggests that using fewer than five levels of random effects may be acceptable in similar circumstances and when one is only interested in estimating fixed effects parameters (*i.e.* predictors, independent variables); in other words, when inference about the variance of random effects terms (*e.g.* sites, individuals, populations) is not of direct interest, but instead are used to group non-independent data. However, there do appear to be some disadvantages to fitting random effects in some of the cases presented here. Coverage of the fixed effects estimates was lower for LMMs when the sample size was lowest (*N* = 30; yet this was true regardless of the number of levels of the random effects terms). Additionally, there were substantially more singular fits with fewer levels of the random effects terms (as well as with smaller sample sizes). Yet, there appear to be minimal differences in coverage and RMSE for models with singular fits (again, excepting some small sample size scenarios).

Those following the "less than five levels" guideline typically fit the blocking factor as fixed effects (instead of random effects), turning a LMM into a LM, which increases the number of parameters estimated by the model (*Clark & Linzer, 2015*). In the simulations, LMMs and LMs did not appear to give drastically different parameter estimates for fixed effects. Often researchers (sometimes nudged by peer-reviewers) cite

this guideline of needing 5 levels before random effects inclusion as a reason why they were unable to use a mixed-effects model (*Bain, Johnson & Jones, 2019*; *Bussmann & Burkhardt-Holm, 2020*; *Evans & Gawlik, 2020*; *Gomes & Goerlitz, 2020*; *Zhao, Johnson-Bice & Roth, 2021*). Although there is confusion over this recommendation, as some opt to use mixed-effects models despite this suggestion (*Latta et al., 2018*; *Fugère, Lostchuck & Chapman, 2020*; *Gomes, Appel & Barber, 2020*; *Allen et al., 2021*), likely because of the numerous advantages that mixed-effects models offer (*Bolker, 2008*; *Kéry & Royle, 2015*; *Harrison et al., 2018*). There may be a trend to follow this rule given that authors or peer-reviewers can easily point out that this rule exists (*Gelman & Hill, 2006*; *Harrison, 2015*; *Kéry & Royle, 2015*; *Harrison et al., 2018*; *Arnqvist, 2020*), but may find it more difficult or time-consuming to make a nuanced argument against following such a rapidly growing rule. I am not proposing a new rule to replace the old one, but rather that we, in the field of ecology, think critically about the tradeoffs between choosing to fit grouping factors as fixed or random effects, and what that means for the estimation of these effects (*Robinson, 1991*; *Gelman, 2005*; *Clark & Linzer, 2015*; *Bell, Fairbrother & Jones, 2019*; *Hamaker & Muthén, 2020*; *Oberpriller, de Souza Leite & Pichler, 2021*).

I hope this work sparks further conversation and debate over this issue. It is important to note that results from simulations, such as this one, are not necessarily generalizable to situations outside the scope of the study (see *Bates et al., 2015a*; *Matuschek et al., 2017*). Instead, it warrants future investigation and further simulation studies with more thorough scenarios. For example, there is much work to be done exploring GLMMs (*Oberpriller, de Souza Leite & Pichler, 2021*), correlations, different magnitudes of fixed effects relative to the variability (signal-to-noise ratio; *Clark & Linzer, 2015*), and variation in the slopes (*i.e.* random slopes; *Matuschek et al., 2017*) and random effect variance. In addition, it would be useful for researchers to simulate data similar to their own research data in order to assess the consequences of using fixed and random effects in each context. Given the widespread accessibility of mixed-effects models, future simulation studies and further assessments of these statistical methods are necessary to understand the consequences of both violating and automatically following methodological rules.

## ACKNOWLEDGEMENTS

I would like to thank Xavier Harrison for inspiration, Trevor Caughlin for aiding in my understanding of simulations over the years, Ben Bolker, Mollie Brooks, and an anonymous reviewer for helpful comments on earlier versions of this manuscript, and Boise State University and the Hatfield Marine Science Center at Oregon State University for general and logistical support.

### Funding

The work was supported by the National Science Foundation (NSF GRFP 2018268606). The funders had no role in study design, data collection and analysis, decision to publish, or preparation of the manuscript.

## Grant Disclosures

The following grant information was disclosed by the authors:
National Science Foundation (NSF GRFP): 2018268606.

## Competing Interests

The authors declare that they have no competing interests.

## Author Contributions

- Dylan G.E. Gomes conceived and designed the experiments, performed the experiments, analyzed the data, prepared figures and/or tables, authored or reviewed drafts of the paper, and approved the final draft.

## Data Availability

All data and code is available at Zenodo: Gomes, Dylan G.E. (2021). Data and code for: Including random effects in statistical models in ecology: fewer than five levels? [Data set]. Zenodo. https://zenodo.org/record/5803559.

## Supplemental Information

Supplemental information for this article can be found online at http://dx.doi.org/10.7717/peerj.12794#supplemental-information.

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
