# Peer review of "Should I use fixed effects or random effects when I have fewer than five levels of a grouping factor in a mixed-effects model?"

_PeerJ, doi:10.7717/peerj.12794_

## Round 0.1 · original submission · Major Revisions

I have received back the comments by three reviewers, who were extremely constructive. I believe this would be a good contribution to the field, but there's still much, much work to be done, as you'll notice. By the tone of the critiques of R1 and R3, I'd make this a reject and resubmit, but the major revisions option in PeerJ has no hard deadline. Despite the strong criticism the manuscript received, I still think it has potential.
Both R1 and R3 expressed concerns about the design of your simulations, which decisively impacted your results making them counterintuitive. Pay closer attention to the issue on p-values raised by R3 and the way LMM vs. GLMM (and N-mixture methods) are treated throughout the text, as pointed out by all three reviewers.

Also, notice that while your paper was in review a preprint with similar objectives, but a rather distinct approach, has appeared

https://www.biorxiv.org/content/10.1101/2021.05.03.442487v1

I'm not sure if this should be cited or acknowledged, but it might surely be useful.

·

Basic reporting

Generally clear and complete throughout.

Experimental design

No comment (see general comments below)

Validity of the findings

Overall this paper makes a good point: I completely agree that it is unwise to thoughtlessly omit a blocking factor from a statistical model if there are 'insufficient' levels. While respecting that a single paper can't do everything, and that one has to start somewhere, I was concerned about the lack of broader theoretical discussion, and the very specific set of simulations on which the results are based.

Additional comments

SCOPE

I am exceedingly nervous about basing general advice on a single set of simulations that can be considered a special case in many ways: (1) a linear mixed model (rather than a GLMM, for example); (2) a balanced design (the values of the numeric covariates X1 and X2 differ across sites, but the number of observations per site, and the range of X1 and X2, is balanced/identical across sites; (3) particular choices for the relative magnitudes of residual and among-group variability, and for the magnitude of the fixed effects relative to the variability (signal-to-noise ratio); (4) evaluating across-site variation only in the intercept (no variation in the slopes either in the simulated data or in the model). I am particularly nervous because of the finding that dropping the random effect [pseudoreplication] doesn't influence the probability of type-1 errors, and the results that show that the rate of rejection of the null hypothesis is independent of subsample size in the real-data case study ... if I got these results, which seem to contradict what would be expected from theory, I would work hard to try to figure out what was special about my setup that drove these counterintuitive results: are the rates of rejection in fact changing in the expected direction, but by a very small magnitude? Is there something about the choice of experimental design and parameters that makes this case unusual, or can these results be reliably generalized to the broad range of ecological data analyses? (For example, Bates et al (2015) criticize Barr et al (2013) for simulations that use unrealistically large effect size, casting doubt on their conclusions.)

COMMON PRACTICE

The author asserts in several place that ecologists drop random effects entirely from the model when they deem that there are insufficiently many levels of the grouping variable (l. 118 "[t]he current state of practice in ecology is to drop the random effects terms"; also l. 333 "typically drop random effects"). Given that it's *not* would I would advise (I would typically say to treat the effect as fixed rather than random, cf l. 192), I'd very much like to know the foundation for the claim: is this actually *recommended* anywhere? Is there a systematic review that evaluates the commonness of the practice? In the (likely) absence of either of these (or other) pieces of evidence, it would be good to qualify this statement by saying that it's the author's observation ... (It would also be good to know if there is published advice explicitly recommending substitution of fixed effects for random ...) (Now I see that there is a potential list of papers supporting this contention on ll. 377-378, and countervailing practice on ll. 379-381; it might be good to foreground this comparison, and perhaps do a slightly more systematic comparison [an analysis of a sample of, say, 50 recent papers from ecology, stating number of levels of the grouping variable and the strategy followed, doesn't sound like an unreasonable amount of work, and would be extremely valuable].

METRICS

The author evaluates bias, finding that the results are effectively unbiased, and type-I error. Bias is OK (it has some odd properties of varying with measurement scale - for example, an unbiased estimator for the variance will be biased for the standard deviation), although I would be more interested in relative root mean squared error (i.e. sqrt(mean((beta_est/beta_true-1)^2))) (I suspect it will still be tiny.) I would strongly encourage the author to evaluate _coverage probability_ (computationally, 90% coverage probability will be easier to evaluate to a given level of precision than 95% coverage, although readers may be more comfortable with 95% coverage). Coverage probability has two major advantages of (1) being equally applicable to zero and non-zero effects and (2) getting away from the morass of what exactly one is doing when evaluating type-1 errors. For example, line 129 needs to be read carefully: if "one does not exist" is read as "when the effect is exactly zero in reality", it's correct, but it's easy to misconstrue as "detect a significant effect when the effect is not significant [but not non-zero]", which is nonsense but superficially plausible. Switching to analyzing coverage, one could also eliminate most of the slightly distracting paragraph after l. 349 that discusses issues with p-values, generally and in mixed models particularly. Finally, type-1 errors *only exist* in simulations, where one can set an effect to exactly zero (ecological effects are practically never zero), but coverage (an estimate of the reliability of estimates of uncertainty) is applicable both in simulations and in the real world. (As long as one is going to compute estimates and confidence intervals for each run, there is little computational cost to evaluating bias, variance, RMSE, and coverage ...)

I might not bother evaluate the estimates of the intercepts (which are rarely of scientific interest) and instead evaluate the scaled RMSE and coverage for a range of slopes.

TERMINOLOGY

Unfortunately the terminology associated with REs is often vague; the difference between the complete random-effect term (e.g. "correlated random intercepts and slopes associated with variation across sites"), the grouping variable associated with that term (e.g. "sites"), the levels of the grouping variables ("site 1, 2, 3 ..."), and the BLUPs/conditional modes determining the predicted deviations of an effect strength [intercept, slope, etc.] in each group from the population mean ("site 1 slope effect = -1.2"), are all sometimes called "random effects".

l. 14: "random effect" = "grouping variable"; "five levels of the grouping variable associated with a random effect" might be more precise.

l. 77 "random effect" = "conditional mode/BLUP"

fixed effect (conservative)
Bayesian (more challenging)
leave alone (overestimate?)


What is suggested for singular models?


Matuschek et al "balancing"




MINOR COMMENTS
====


l. 50-51: occupancy, N-mixture, mark-recapture models don't really belong (IMO) to the class of GLMMs: most of them require a more general form of latent variable than GLMMs can accomodate. (They are "hierarchical" or "latent-variable" models, but not GLMMs ...)

l. 56: I would call binomial, Poisson, etc. a "response distribution" or a "conditional distribution" rather than a "data generating process"

l. 57-58 the "sampling distribution" refers to the distribution of an estimated coefficient across realizations, not to the conditional distribution of a response variable.

l. 58 not sure what "mean-centered" means here. Even more picky, technically a Gaussian would be most appropriate for a *real-valued* response (i.e. one where negative values would be realistic); non-negative or positive values as given in the examples would better use a Gamma or log-Normal (or other transformed-Normal) response distribution. That said, if the coefficient of variation is small, then modeling these variables as Gaussian is standard.

l. 65 why are "average temperature" and "sound pressure levels" italicized?

l. 66 "other" variables are often referred to as "nuisance" variables in statistical contexts (this is used below on l. 116; if it were defined here one wouldn't need quotation marks later)

l. 79 "This": unclear antecedent

l. 97 I might say something about "documented" or "systematic"; there's presumably a lot of _anecdotal_ empirical evidence developed by the authors quoted ... even if some are merely copying assertions from previous authors ...

l. 107 why emphasize "most cases" when "many cases" is sufficient (and Kéry and Royle already say "vast majority")?

l. 124 "this": unclear antecedent

Figure 2 caption "The bias in sigma starts to approach the starting (simulated) sigma=0.5" should "bias approaches zero" or "mean estimated value approaches the true value"?)

eq 2.1 it is more common to include the overall intercept (mu) as a constant in the equation for the response, and make the site-level effects explicitly zero-centered (i.e. Normal(0,sigma)) (I prefer a centered dot (2·b), juxtaposition (2b), or perhaps an x (2×b), for multiplication, rather than *, in mathematical equations)

l. 207 "latter of which" (awkward)

l. 214 I would expect that a binomial confidence interval (e.g. Clopper-Pearson, although the Normal approximation should be OK with p~0.05 and 10000 samples) derived from the number of samples (10,000) and the observed probability would be fine - bootstrapping seems overkill.

l. 248, 256 "resurrect" doesn't seem quite right; "reconstruct"?

l. 253 "lead" → "led" or "eads"

l. 256 "non-important" → "unimportant"

l. 281 "probability of"

l. 337 "higher probability of significant p-values" → higher probability of rejecting the null hypothesis at alpha=0.05

===
Barr, Dale J., Roger Levy, Christoph Scheepers, and Harry J. Tily. “Random Effects Structure for Confirmatory Hypothesis Testing: Keep It Maximal.” Journal of Memory and Language 68, no. 3 (April 2013): 255–78. https://doi.org/10.1016/j.jml.2012.11.001.

Bates, Douglas, Reinhold Kliegl, Shravan Vasishth, and Harald Baayen. “Parsimonious Mixed Models.” ArXiv:1506.04967 [Stat], June 16, 2015. http://arxiv.org/abs/1506.04967.

Matuschek, Hannes, Reinhold Kliegl, Shravan Vasishth, Harald Baayen, and Douglas Bates. “Balancing Type I Error and Power in Linear Mixed Models.” Journal of Memory and Language 94 (June 1, 2017): 305–15. https://doi.org/10.1016/j.jml.2017.01.001.

·

Basic reporting

The paper is mostly well written. Only a few minor changes are suggested below.

On the literature, see Clark & Linzer 2015 (this paper should probably be discussed in the intro and discussion as it is quite similar).

Clark, T., & Linzer, D. (2015). "Should I Use Fixed or Random Effects?" Political Science Research and Methods, 3(2), 399-408. doi:10.1017/psrm.2014.32)

Experimental design

This paper addresses an interesting topic in statistical ecology using both a simulation study and case study. Overall, the experimental design is good.

L 262-264: Consider using REML to get more accurate variance estimates.

Validity of the findings

The premise of this article was not clear until line 196 and many statements make it unclear throughout. It sounds as if they were going to drop the effect of site rather than include it as a fixed effect. In several places, it talks about dropping (L 118-120, L333), ignoring (L 207-208), missing (L 365) or omitting (L 369) the random effect which is not exactly the case in the LMs. The abstract even says that the LMs commit pseudoreplication (L22-24) but this is false because they include a fixed effect of site. When the literature says that fewer than 5 levels of a random effect are too few, the alternative is to account for it using a fixed effect as was done here.

Additional comments

Results of simulations should state how often variance was estimated as zero (to some precision) or non-convergence warnings/errors were produced as these are some of the reasons that people suggest > 5 levels of a random effect.

L 18: "terms" should be "levels"

L43: The citation for glmmTMB should be
> citation("glmmTMB")

To cite glmmTMB in publications use:

Mollie E. Brooks, Kasper Kristensen, Koen J. van Benthem, Arni
Magnusson, Casper W. Berg, Anders Nielsen, Hans J. Skaug, Martin
Maechler and Benjamin M. Bolker (2017). glmmTMB Balances Speed and
Flexibility Among Packages for Zero-inflated Generalized Linear Mixed
Modeling. The R Journal, 9(2), 378-400.

A BibTeX entry for LaTeX users is

@Article{,
author = {Mollie E. Brooks and Kasper Kristensen and Koen J. {van Benthem} and Arni Magnusson and Casper W. Berg and Anders Nielsen and Hans J. Skaug and Martin Maechler and Benjamin M. Bolker},
title = {{glmmTMB} Balances Speed and Flexibility Among Packages for Zero-inflated Generalized Linear Mixed Modeling},
year = {2017},
journal = {The R Journal},
url = {https://journal.r-project.org/archive/2017/RJ-2017-066/index.html},
pages = {378--400},
volume = {9},
number = {2},
}

L45: missing comma after "biologists"

L48-51: Kéry has changed the way he refers to site-ocupancy and N-mixture models since his book which equated them with GLMMs; now he calls them "non-GLMM hierarchical models" because they're slightly different from GLMMs. This text needs to be changed accordingly.

L57: Warton 2005 found that negative binomial is usually good for abundance data, so it would be better than Poisson when you reference wildlife counts.

Warton, D.I. (2005), "Many zeros does not mean zero inflation: comparing the goodness-of-fit of parametric models to multivariate abundance data". Environmetrics, 16: 275-289. https://doi.org/10.1002/env.702

L 68-70: I don't think this sentence is necessary and could draw criticism because it only applies to GLMMs in an MLE framework, but not a Bayesian one.

L 79 and elsewhere: "group" should probably be "level of the random effect" or "blocks" or make a statement that you're using it this way throughout because the meaning could be anything in line 84. This is sort of done on line 96, but could be earlier and more direct. In the methods section around line 148, this changes to "site" for a site-level intercept.

L 224: Give original data set size before resampling scheme.

L 368 and throughthout "repeat-measures" sounds odd to me. Isn't it usually "repeated-measures".

Reviewer 3 ·

Basic reporting

The manuscript focuses on the use of Mixed-effects models in ecology, specifically their use to account for non-independent observations due to grouping of e.g. sites. It is well written, in a way that is very accessible to most ecologists. The author focuses on the “5-level” rule, where researchers have a strong focus on the number of levels required to estimate the variance parameter(s) of a random-effect. Specifically, the author focuses on the case of a random intercept, with a balanced designed and two predictors. Though Generalized Linear Mixed-models are mentioned on various places, the manuscript only focuses on (General) Linear Models (LMs) and Linear mixed-effects models (LMMs). The manuscript could be significantly improved by expanding the simulations to include GLMMs, though I understand that requires significantly more work. Alternatively, the author can keep focus on LMMs, but then please remove the references to GLMMs entirely, as results are likely to differ depending on the chosen distribution.

Unfortunately, the text includes a range of strong, and/or subjective, statements without references, that needs to be tempered. This strong language use is unnecessarily off-putting, and makes it harder to convince a reader of the, in my opinion, valid argument of treating a rule-of-thumb as a rule instead (in addition to it being mildly annoying).

The figure legends should be revised, to only include an explanation of a figure, rather than including results and/or discussion of those results.

Experimental design

The author fits linear mixed-effects model, and retrieves p-values, to explore if misspecification of the model (i.e. no random-effect when it is required), increases Type-I error. However, in various places in the manuscript (e.g. line 281 and figure 4) the author misinterprets the p-value as (for example) “probability of the null hypothesis”. A p-value is always accompanied by a test-statistic, though the author does not state anywhere in the manuscript which test is used (in the lmerTest R-package), so that the paper focuses on p-values, rather than on the test that the p-value is a result of. This is a mistake many ecologists make: please correct this throughout the manuscript. Additionally, note that a p-value is correctly interpreted as “the probability to get this test-statistic or more extreme, if the null hypothesis is correct”.

The lmerTest package provides methods to calculate degrees of freedom for the F-distribution, in a specific manner not supported by the developers of the lme4 R-package, for mixed-effects models with unbalanced designs. This discussion, for how the degrees of freedom of the test-statistic should be calculated, differs for LMMs and GLMMs. I refer the author to the excellent texts that Ben Bolker has written on this subject, e.g.: https://bbolker.github.io/mixedmodels-misc/glmmFAQ.html#why-doesnt-lme4-display-denominator-degrees-of-freedomp-values-what-other-options-do-i-have. As such, my suggestion to make this paper more interesting and relevant, is to include an unbalanced design in the simulations. Again, to summarize, it is a known result is that the fixed-effects estimates will be unbiased when the model is misspecified and a random-effect is omitted (and as a result, so are the test-statistic and the accompanying p-values
For a paper that attempts to explain the difference in fixed- and random-effects modeling, the author surprisingly rarely mentions technical details that are important to understanding that difference. For example, for fixed-effects the likelihood is maximized. Whereas when including random-effects, the joint likelihood is marginalized w.r.t. the random effect, while maximizing with respect to all parameters. Please elaborate in the text.

The author includes a case study to appeal to the applied nature of the target audience. However, the procedure used is overly complicated and doesn’t address some of the issues that arise due to the hierarchical nature of the data, in a non-parametric bootstrap. In various places the author states that “the true values of the parameters are unknown”, which takes away any relevance that this section has to the manuscript. I make suggestions in the detailed comments below on how this could be improved.

Validity of the findings

The author extensively presents the issue with p-values in mixed-effects models in the discussion from an ecological perspective, but unfortunately fails to clearly identify and discuss the problems it has. In a balanced design, such as here, with normally distributed residuals such as here, it is well known that the test-statistic is exactly F-distributed with known degrees of freedom, so that it nullifies any discussion around p-values in LMMs. Unfortunately, the author has focused completely on a balanced design, but fails to realize the estimates of a linear regression are unbiased when the random-effect is omitted, in such cases. There is plentiful statistical literature available from the last 50 years, or so, on mixed-effects models and balanced designs, though the author does not seem to be aware of this. If the author insists on keeping the manuscript focused on a balanced design, they should check that statistical literature for a deeper discussion on the matter, to provide a stronger justification here.

Additional comments

Next follow some line-by-line detailed comments, though I suggest to thoroughly revise the whole manuscript. In general, the paper reads as if it lacks statistical sophistication. While I appreciate that it should be accessible to applied researchers, I suggest improving the text so it does not appear as if the subject has been “dumbed down”, as it is inherently a complex and technical subject that the author is attempting to address that statisticians have struggled with for decades.

Detailed comments
Line 12: remove generalized, or include GLMMs in the study design. This “oversells” what the manuscript is about.
Line 17: Remove ecological from “simulate ecological datasets”. There is nothing ecological about a simulation, though it might be a realistic representation of ecological datasets (if so, write that).
Line 19: remove should and similar strong language use throughout the manuscript
Line 23: correct generalized to general.
Line 25: please make sure to use correct language use. Correct “LM uncertainty” to “uncertainty in parameter estimates of LMs” or similar.
Line 26: remove never or provide a range of references that prove this.
Line 31-32: please remove this line “blindly following guidelines” or provide various references that prove that this is what ecologists do (rather than what the author thinks ecologists do)
Line 38-43: this is a very long and difficult to read sentence, please improve.
Line 46: I doubt that both Bolker references are required here, one should be sufficient.
Line 49-53: this manuscript has little to do with accounting for imperfect detection, N-mixture models of mark-recapture methods so I fail to see the relevance of this paragraph. Please remove.
Line 55: remove “a regression type analysis", this language use is too informal and redundant
Line 56: please correct “data generating processes” to “distributions”. The data generating process envelops both the model and the distribution, while here the author only refers to the distribution.
Line 58: remove “mean-centered”, this is not a requirement for linear regression, nor is it usual to do so.
Line 60: remove “simply”
Line 67: the author their argument for random-effects is that they account for nuisance variation, using fixed-effects to account for unexplained variation here, is then misspecification of the model (as the author defines them as random-effects).
Line 76: replace “should” with less strong language
Line 79-81: the author argues, without any references or clarification, that modelers should use random slopes. This is context dependent, and this is a too general statement. Please remove, or improve by clarifying considerably (I suggest the former).
Line 97: please provide some more/other references than Harrison. A large part of the author their argument (also further on in the text) is build on Harrison 2015, while they argue that it is a problem that has wide scope and evidence, so using a larger variety of references to show that shouldn’t be an issue.
Line 102: please provide some references on this matter, as this is again a very bold statement.
Line 107: also referred to as “practical identifiability”, see e.g. Wieland et al. 2021 titled “On structural and practical identifiability”.
Line 107: the author in various places argues in a certain way, but fails to see that it is straightforward to argue opposite. Here: there are plenty of cases where residual (random-effect) variances are interesting to inference. Arguing in such a way distract from the novelty of the manuscript so that it fails to convince the reader. Please improve.
Line 118-120: again, the author argues that there is certain practice in ecology, for the application of models, but fails to provide a reference. A lot of these statements seem to be built on conjecture. Please improve.
Line 121: I think you mentioned this earlier, but this isn't about pseudoreplication: that is being controlled for whether a fixed or random effect is used regardless.
Line 146: (equation 2): for a normal distribution where the mean is assumed to be zero (as in all random-effects, because it is common practice to reparametrize as such for the sake of convergence) please write a zero, not a mu symbol.
Line 150: remove mu.
Line 152: this applies to the whole section. I suggest adapting the simulation so that the variance of all terms sum to one, to that their contribution can be directly interpreted in terms of R^2.
Line 160: this means that the variance of the residual is (very) small, so that simulated datasets are (very) simple, and the contribution of fixed- and random-effects provide (very) good fits. It makes the whole simulation set-up not very convincing. Please increase variance explained by the random-effects and in the residuals. Similarly, for variances close to zero the normal distribution becomes very “spiky” so that things can get a bit wonky numerically.
Line 185: please remove #LMM:, as it is redundant. Similar for the second code block. I would also suggest not assigning a different model to “m1” and “m2”, but to uniquely name them, as they are different models. Further on in the manuscript this has been done correctly.
Line 190-191: this sentence reads a bit awkward, please improve. We don’t “fit sites” in ecology. We fit a model to our data.
Line 192: add is before if and remove is after the comma
Line 203: please add to clarify that these are still intercepts, though. The variance of the random-effect can as such still be accounted for, by setting the sample variance of the vector of intercepts to the variance of the random-effect.
Line 204: remove this fixed-effects explanation. This is explained earlier in the manuscript and thus is here redundant.
Line 206: in various places in the manuscript the author emphasizes large numbers of the simulation as if they themselves are impressed by it. I would like to note that these kinds of numbers are common for statistical simulations, and the reader is not interested in reading them all throughout the manuscript. Though necessary to report the number of simulations, if the author really wants to emphasize other of these numbers, please do so in a single place. Better yet, just leave them out as it is very distracting.
Line 211-216: I fail to understand why the author is re-sampling p-values from simulated models. Instead of making this more complicated than it has to be, the author can draw a histogram of p-values from the simulated model, which should follow a uniform distribution, with 5% of p-values being smaller than 0.05. Note, that for a single trial, the p-values will follow a Bernoulli distribution with p=0.05, if the author wants to perform a formal test of some kind.
Line 218: the author clearly wants to simulate a dataset realistic for ecological purposes, but this entire case study fails to convince as the author themselves emphasize, on multiple instances, “the truth is not known”, making this whole section feel redundant. My suggestion is to do this as is common in statistics: fit a model to the data with the desired terms, predict using that model, and consider the first fitted model as the truth. That way, the simulation is ecologically realistic, and the true parameters are known. In essence, please revise this whole paragraph/section.
Line 234: the linear model is not in “base”, it is in “stats” which is distributed with the core of R, just as “base”. It is important to get the details right, this is very sloppy.
Line 248: various instances the author uses the word “resurrect”. It’s meaning in this context is not clear to me, please swap with something more appropriate.
Line 254: please be clear, this “uncertainty” is variety in the simulation. This helps for inexperienced readers that might confuse this with the standard error retrieved from a model object. If the target audience is applied ecologists, it is important to describe these details as clearly as possible. Currently, that is not the case.
Line 261: correct mean to intercept.
Line 263: the author uses the word “bias” on various places throughout the manuscript, but never actually calculates e.g. bias, or Root mean squared error, or some other metric. Please correct language use or include such a calculation, so that bias is actually presented.
Line 255: the percentages are also numbers I (personally) don’t care for. I can read these things in the figure, as the actual decimals are (truly) irrelevant to the arguments made here. Please remove, overall the manuscript will become a lot cleaner, neater, and in general easier and more enjoyable to read.
Line 265: again, talking of bias but providing percentages rather than bias. Please correct. Also, how important would this bias be, if we would be making inference on the variance parameters, which is one of the main points the author makes: if we don’t infer on the variance parameters everything is fine. So what if we do? Is there really any problem?
Line 268: again, these numbers are not interesting to me. If the author wants to present bias, then present a bias metric, not a percentage.
Line 280: be clear which type-1 error rate is meant here. Type-1 error rate of the test-statistic for the fixed-effects slopes.
Line 281: probability type-1 error? It’s unacceptable for a manuscript on p-values to contain mistakes in the presentation of the meaning of p-values, make sure this is correct throughout the manuscript.
Line 285: this whole section is redundant at the moment because of how this has been set up. Please read my previous comment on this and revise.
Line 293: remove 10.000
Line 303: this sentence doesn’t read right. The levels of an accompanying random effect have fewer than five levels?
Line 304: what was your expectation for the type-1 error rates? It would be helpful if that was included in the end of the introduction.
Line 319: adding the mu and sigma symbols doesn’t add anything here, please remove. Same for the next line.
Line 324: remove quite
Line 330: how often do ecologists report variance parameters of random-effects? Is this really an issue? The author has failed to convince me on this (and some other) fronts.
Line 337: a higher probability of p-values? I think you mean proportion here.
Line 347: many readers might be put off by such statements, as the paper might seem (too) subjective, please improve.
Line 349-362: remove this whole paragraph. The author has clearly failed to see the issue with the lmerTest R-package and needs to consider his issue more deeply. By using the R-package the author does send the message that it is OK to do this, whereas they seem to argue that it isn’t. Confusing.
Line 388-389: remove this sentence. I really think a more objective view of this subject is required in this text.
Figure legend 4: please refrain from drawing inference oor conclusions in figure legends. (and reduce the size of the figure legend). Here, the author just needs to explain what is presented. The reader can, and should, interpret this on their own, given that sufficient information is provided on the context of the figure

---

## Round 0.2 · Minor Revisions

Thank you for resubmitting the revised version. While R2 makes only a few minor comments, R3 noticed that some important changes still have to be made.

Please, implement those changes and submit a new version with a rebuttal letter.

·

Basic reporting

Basically OK.

MINOR COMMENTS

l. 14 "[so] few levels"

l. 20 "[the] coverage probability"

l. 24 "simulations/model structures" → "scenarios"

l. 30 "blindly" → "routinely" ("blindly" is evocative, but seems too strong. "Automatically" ?)

l. 64 "date and time of sampling" are probably not the best examples of fixed-effect nuisance variables, as both could easily also be treated as categorical grouping variables in a random-effect term. Size, age, or sex of an individual organism would be more typical/less ambiguous (admittedly I can't immediately think of good parallels in a climate change or anthropogenic noise context).

l. 72 "usually" (stronger than "often" - in reality it's "almost always")

l. 73 "which is not the same distribution" is vague: does it refer to the "data-generating process", which is not a distribution (and hence can't be "the same as" the response distribution) ? Do you mean the conditional distribution?

l. 94, 96; I found it interesting that the author has contributed to studies that both do and don't follow the ">5 levels" rule ...

l. 103 "In many cases,"

l. 114-15 "in ecology [when the grouping variable has too few levels] is ... rather than [as] random effects"

l. 115 "and" splice (split sentence or use a semicolon?)

l. 123 (and 226, 261) I have strong feelings about the use of "warnings" vs "errors", which have specific technical meanings in R. Singular fits in lme4 lead to *messages* saying that the fit is singular. Fits with other packages that estimate RE variances on the log scale typically lead to *warnings* about convergence. Perhaps just "singular fits" ?

l. 128 Please specify the lme4 package version used.

l. 143 It seems odd to use a model with intercept equal to zero. There's nothing statistically wrong with it, but if we're thinking about the response variable as "tree height" then it's weird. You can always say "it's mean-corrected tree height", but that seems to introduce an extra step. (If I'm not mistake the response to reviewers stated "Yet, it is my understanding that one needs a response with both negative and positive values for linear regression", which is plain wrong ...)

l. 147 "Note that" → "While" ?


l. 163-4 "a [strong] positive relationship ... weak [positive] relationship" (is there a biological idea behind the second relationship? A trophic cascade?)

l. 170 Using both back-ticks and typewriter font to refer to package names seems redundant (back-ticks should probably only be used in very specific contexts where users are used to "markdown" conventions)

l. 173-174 it's not clear why both forms of the equation are included here. I can appreciate that the author wants to convey both the specific and general forms of the question, but this approach seems clunky.


l. 211 "centered on " → "close to" (you can say either that the distribution of estimates is centered on the true value, *or* that the means are close to the true value ...)

l. 214-216 it would be good to include some typical numerical values in the text here so the reader isn't forced to look at the figure to know approximately what "high" and "low" mean.

l. 230 missing space "that fixed"

l. 253 arguably *any* rule of thumb/cutoff for how many levels is enough will be arbitrary???

l. 258 "there do appear to be some disadvantages"

l. 262 "sample sizes"

l. 315 This is the wrong citation for Bates et al.; please cite the JSS paper (e.g. 'citation("lme4")' in R)

Table 1. There are few enough distinct categories here that a graphical presentation might be nice. I'm thinking about something like

1. reorder "random_effect" variable by mean nlevels (using reorder())

2. ggplot(data, aes (x=nlevels, y = random_effect)) + stat_sum() +
scale_x_log10()

(stat_sum() will be useful for distinguishing overlapping points (there are two studies with {sites, 6}))

Figure 1. It would be useful either to put confidence-interval ribbons on the individual lines, or to put a CI around the expected 0.95 value. (I think this will be approx ± 2*sqrt(p*(1-p)/n) = ± 0.004, which actually seems surprisingly small; if the sampling error is this small, it makes me wonder about apparent artifacts like the non-monotonic pattern for LMM N=60?)


I rarely find tables of numbers useful, I always prefer graphical summaries.

Figure 2. It's impossible to distinguish the line type in the legend.

Figure 3. Are the confidence intervals shown the *means* of the CIs computed for each simulated data set (i.e. the means of the lower and upper limits), or ... ??

Experimental design

l. 173-174 The formulas here are wrong (i.e., don't match the model described in the text/on l. 187) because R implicitly includes an intercept in the model unless +0 or -1 is included in the formula.


l. 155 it's interesting that the author chose to simulate a balanced, nested design, i.e. a design where the predictor variables vary only across levels of the grouping variables (sites), not within sites, and where there is an equal number of observations per site. As pointed out by Murtaugh (2007), this is a case where it's arguably better (given that we're not interested in quantifying variance) to collapse the data by taking the mean tree height in each site, at which point the model collapses to a single level and we don't even need mixed models!

Murtaugh, Paul A. “Simplicity and Complexity in Ecological Data Analysis.” Ecology 88, no. 1 (2007): 56–62.

Validity of the findings

The revised version of this paper is considerably improved. I still think the pros and cons of 'dropping' random effects (i.e. coding a grouping variable as fixed rather than random) could be stated more clearly; that the scope of the simulations is rather narrow; and that the paper could use a more solid theoretical foundation.

1. The primary thing missing from this paper is a clear, explicit statement of the advantages and disadvantages of the two approaches (random vs fixed grouping variables). For example:

* LMs are simpler to use (and hence less prone to user error), and faster to compute.
* Because they are conceptually simpler, metrics like R^2 are easier to compute.
* They avoid concerns about singular fits.
* They preclude users from making possibly inappropriate generalizations to unobserved levels of the grouping variable.
* They eschew assumptions about the distribution of the random effects.

* LMMs are (often) conceptually correct models of the system.
* They allow generalization to unobserved levels of the grouping variable.
* In principle LMMs can provide RMSE and increased precision (I don't see an analysis of the relative efficiency/width of confidence intervals for LM vs LMM, which would be useful; Fig 2 shows that over this range of scenarios the RMSE advantage is weak at best).
* In a study with multiple analyses with widely varying numbers of grouping variable levels, it would be more consistent to use LMMs throughout rather than switching from LMMs to LMs at some threshold value.

2. It would be extremely useful to do a subgroup analysis that compares the singular and non-singular fits. If the results from singular fits are not inappropriate (biased, undercovered, etc.), that would be an important guide to users to know if they should worry when they get a singular fit.

3. Please give at least a brief description of the conclusions of the "limited" previous studies that have tackled related questions (i.e. Oberpriller et al., Clark & Linzer, Gelman & Hill, Bolker 2009).

Figure 1. It is strange, and very concerning, that the coverage for the 'weak' slope effect should *decrease* as the sample size increases. This seems extremely hard to explain; when the statistical model matches the simulation model, the coverage should converge to the nominal level as the sample size increases. Please double-check the code/results, *or* develop and present a theoretical explanation as to why this pattern emerges.

Reviewer 3 ·

Basic reporting

The author has done well in addressing the many detailed comments, especially related to strong language in the manuscript. Largely, the author has removed any reference to p-values and has a such addressed most comments related to that subject. There was a case study included in the previous version of the article, which was also removed in response to a comment. As such, the article now fully focuses on a simulation study for the assessment of bias in fixed-effects parameter estimates in the presence of a random intercept term, with an increasing number of levels.
There are still some minor textual errors that need addressing, and I have two larger comments which need careful consideration.

Experimental design

-

Validity of the findings

One comment from my previous response has not been addressed to satisfaction. This relates to the difference of mixed models for use of balanced versus unbalanced design studies. My comment on the original submission related to the inclusion of p-values, their application in mixed-effects models, and the asymptotic distribution of the test statistic under a balanced design. Though the author addressed the difficulty of p-values in mixed-models by removing large parts of the article, the author still seems unaware of the deeper issue that underlies that discussion. Since the author explicitly decided to not expand the article by including an unbalanced design, this is now an issue more central to the whole article. For clarity, I refer the author to the classical text of Searle (1997), but discussion of this topic can also be found in more recent textbooks such as Wood (2017), or in general in statistical textbooks on linear modeling. It is a known result, that in the case of a balanced design, the fixed-effects parameter estimates are independent of the estimator for the residual variance. If one adds any random-effects terms, such as a random-intercept, this remains the case. For example, Wood (2017) writes on page 74: “hence, in a mixed model context, we can still use fixed effects least squares methods for inferences about any fixed effects whose estimators are independent of all the random effects in the model”. As such, I largely suspect that any deviation from that known result in this article is due to the software implementation. The author can address this by altering the study design to that of an unbalanced design, e.g., by setting the number of observations for half of the levels in the random effect to fewer observations (perhaps 2 or 3).

My second comment applies to the removal of the ecological example. In the previous version of the article, the example did not add (much), as the author questioned the usefulness of the example in the text, by stating e.g., “the truth is not known”. Since the target audience of this article are applied ecologists, and the author has explicitly stated the interest of conveying this complex subject in an understandable way, I urge the author to re-implement the ecological example, but to improve it compared to the original submission. The example has the potential of making the subject a lot more understandable to most readers, though it does need to be designed as such. For example, the author could instead discuss how following the 5-level rule might change inference in a real data example, to convince the reader of the issue from a more practical perspective. In case the inference is the same, then perhaps address the issues of numerical stability of the fitting process.

Please note that there are no means (referred to as mu in the manuscript) for the random-effects, as the random-effect is assumed to have a mean of zero (as stated in eq 1). I suppose the author intends to refer to the best linear unbiased prediction (BLUP) of the random effect, which also relates to my comment on line 2 below.

Lastly, I had trouble connecting the figures to the text, as the images were not named or numbered in any way so that they could easily be related to the text.

Below follow more detailed comments to the text, though I suggest to first consider thoroughly revising the manuscript based on these main comments.

References:
Searle, S. R., & Gruber, M. H. (2016). Linear models. John Wiley & Sons.
Wood, S. N. (2017). Generalized additive models: an introduction with R. CRC press.

Additional comments

Detailed comments:
Line 2: The author has chosen the title to include the terms “fixed” and “random” effect. Yet chooses not to further expand on the details between the differences (this was one of my comments on the previous version). I urge the author to re-consider: the difference between fixed- and random-effects is often unclear to ecologists. Yes, it is a complex and detailed issue, but at least the author can mention it in a technical context (maximizing vs. marginalizing) and provide the reader the option to continue reading on that elsewhere.
Line 14: remove such, the whole point is that at 4,6 or 7 levels the estimation of the random-effect could also be difficult: it’s a rule of thumb.
Line 18: replace “and” by “to” and remove “too”
Line 20: e.g., should be i.e.,
Line 20: change “although” to “the”
Line 21: remove “in some cases”, “may” before in the sentence is sufficient to indicate speculation
Line 23: change in brackets to “when they are nuisance parameters..”
Line 24: change forward slash to “and”
Line 48: change “model correlations” to “account for correlation between”
Line 49: I have not seen random-effects referred to before as “varying effects”. Is this addition necessary, I don’t think it is widely referred to as such in ecology?
Line 55: it is unclear to me what is meant by “body condition residuals”. Potentially confusing language use with “residuals” in the context of a regression
Line 58: should mixed effects be italized?
Line 64: I still don’t particularly agree with this explanation. Date or time of sampling is one of those examples that can very well be understood as a random effect. Is there a more clearly distinguishable example the author can come up with, that is (essentially) never used as a random effect? Habitat perhaps?
Line 66: I don’t follow the author their meaning with this sentence
Line 70: the sentence after the reference seems disconnected with the previous, but they also seem to convey the same message. Is it possible to improve this?
Line 71: “Random-effects are random” seems to be a (too) obvious statement. Please change to “Random-effects are assumed to be drawn…”
Line 86: change “this” to “the”
Line 88: missing a word “too few draws from distribution”
Line 89: add “at least” before five
Line 95: why do they ignore the rule if they acknowledge its existence? Can you provide context to this statement? It seems relevant to your argument.
Line 98: I’m confused about table 1, only one study included a random effect with fewer than 5 levels, is that correct? Also, why does the table have 27 rows if there were 29 studies with random effects?
Line 103: multiple random effects implies multiple variance parameters, yet variance here is singular not plural. Do you mean a single random-effect with multiple levels?
Line 110: change “it is” to “can be”
Line 113: uncertainty of predictor variables is a different problem entirely (error-in-variables). I think you mean uncertainty of parameter estimates.
Line 114: we don’t “fit blocking factors”. We treat them as fixed-effects rather than as random-effects, please correct.
Line 129: add “the” before “R package”
Line 137: eq 1 and further lack the explicit normal distribution for the residual, epsilon. Also, can we get the equation numbers on the same lines as the equation?
Line 149: relates to a previous comment with respect to R^2 interpretation. You can increase the magnitude of the variances, and still interpret proportionally. They don’t need to all sum to one.
Line 154: please improve so that this is a correct and full sentence. Also, equations are part of sentences and need a full stop (or comma) after them consequently
Line 157: it is more common to parameterize the normal distribution with variance, i.e., sigma^2.
Line 159: I don’t think you defined subscript j before
Line 162: space missing after the second brackets. Also, sigma is usually referred to as the standard deviation, not the variance.
Line 170: remove quotes around R-package names
Line 180: I think should be n here (total number of levels)
Line 189: beta is a parameter, not a term
Line 213: extra space after which
Line 215: I would never consider slopes “strong”. First, this is a bit subjective language use, second, the slope represents the effect of the predictor and has no effect in and of itself.
Line 215: please refrain from similar subjective language “high”. I would suggest changing to “higher” as its relative to LMMs
Line 217: I miss the general conclusion that coverage probability was lower for LMMs than for LMs
Line 218: RMSE of fixed-effects parameter estimates, I think? Please clarify
Line 220: Please add confidence interval bands to all figures, otherwise it’s difficult to draw such conclusions
Line 220: change “strongest” to “largest”
Line 222: The random effect has a mean of zero, I think you mean the BLUP here too? Please see my other comments on this before
Line 230: typo
Line 231: remove (n>5), it doesn’t add anything here
Line 232: extra space after level
Line 232: please improve “random effect sigma” to “random-effects variance” or what you actually refer to here
Line 248: there are no mu’s in the random-effect (well, they’re zero) please clarify (see also comments before)
Line 265: please note that specifying a random-effect as fixed-effect as here increases the number of parameters in the model
Line 279: change “as researchers” to “field”
Line 280: again, this requires consideration of marginalization vs. maximization

---

## Round 0.3 · accepted · Accept

Thank you for resubmitting the revised version along with the responses to reviewers’ comments. I can say that this is a much-improved version as compared to the first one. I am satisfied with your responses and also with the way you implemented the modifications in the text and I’m happy to recommend acceptance.

The Section Editor commented that the title could be more clear and descriptive if the author wants to encourage readership.